# Accurate prediction of flux distributions compatible with metabolite concentration effects in genome-scale metabolic networks

Fayaz Soleymani[1,2], Zahra Razaghi-Moghadam[2], Zoran Nikoloski[1,2]*

**1** Systems Biology and Mathematical Modeling Group, Max Planck Institute of Molecular Plant Physiology, Potsdam, Germany, **2** Bioinformatics Department, Institute of Biochemistry and Biology, University of Potsdam, Potsdam, Germany

\* nikoloski@mpimp-golm.mpg.de, zoran.nikoloski@uni-postdam.de

## Abstract

Intracellular fluxes shape all cellular functions, and understanding how they are shaped by the joint effects of enzyme abundances and metabolite concentrations *in vivo* currently requires gathering matched quantitative proteomic and metabolomic data sets from resource-intensive experiments. Here, we present KineFlux, a hybrid approach that combines machine learning with enzyme-constrained metabolic models to accurately predict steady-state flux distributions using only quantitative proteomic data. KineFlux builds machine learning models for metabolite concentration effects on reaction fluxes, obtained by using fluxomics and proteomics data from a training set of experiments. Using fluxomic and proteomic data sets of *Escherichia coli* and *Saccharomyces cerevisiae*, we show that the steady-state flux distributions predicted by KineFlux are in line with fluxes estimated by classical approaches. We also demonstrate that the machine learning models embedded in KineFlux are transferrable at marginal loss of accuracy using independent testing data from *E. coli*. Therefore, KineFlux expands the usability of enzyme-constrained models towards accurate prediction of genome-scale flux distributions compatible with metabolite concentration effects without knowledge of enzyme kinetics.

## Author summary

Although intracellular fluxes shape the physiology of every organism, we still lack approaches for their accurate, high-throughput estimation. Here we show that a hybrid approach, that combines machine learning with enzyme-constrained metabolic models, can effectively address this challenge and allow accurate prediction of intracellular fluxes at a genome scale across diverse experimental scenarios with usage of proteomics data alone. The hybrid approach relies on deriving metabolite concentration effects from a training set of fluxomic and proteomic data, and uses machine learning models to predict these effects in a

**Data availability statement:** All data and code necessary for reproducing the findings can be found at: https://github.com/fayazsoleymani/KineFlux.git.

**Funding:** F.S.B. was supported by the German Research Foundation (DFG), Project number NI 1472/16-1 (to Z.N.). Z.N. would like to thank the funding from the NovoNordisk Foundation, grant number NNF23OC0085412. The funders had no role in study design, data collection and analysis, decision to publish, or preparation of the manuscript.

**Competing interests:** The authors have declared that no competing interests exist.

transferrable and interpretable fashion. The hybrid approach expands the applicability of enzyme-constrained metabolic models that are becoming available across diverse species.

## Introduction

Reaction fluxes characterize the integrated outcome of transcription and translation and determine all metabolic functions that support survival, reproduction, and fitness of organisms. Availability of data on intracellular fluxes alongside with metabolomic and proteomic data have facilitated the estimation of enzyme parameters [1–6], the elucidation of metabolites acting as regulators [7,8], and prediction of other physiological traits [9]. However, intracellular fluxes are considered an ephemeral phenotype since they cannot be directly measured. Therefore, considerable efforts have focused on estimating or predicting intracellular fluxes using different data and computational approaches.

Fluxes can be estimated from measurement of isotope label incorporation into different metabolic pools using approaches from metabolic flux analysis [10–12]. However, these approaches entail considerable experimental resources and tedious experiments that must obey the modeling assumptions. Constraint-based modeling offers an alternative and provides approaches for flux prediction by imposing different biochemical constraints that any biological system must respect [13]. Together with principles of parsimony and optimality, these constraint-based approaches have been shown to lead to predictions in line with estimates from flux metabolic analysis based on the same metabolic models [14,15].

Recent advances have resulted in enzyme-constrained metabolic models that allow prediction of fluxes in line with protein constraints and respecting enzyme turnover numbers [6]. These enzyme-constrained models have led to more accurate predictions of maximum specific growth rates on different carbon sources [16–18], flux distributions [18], and other phenotypes [19,20] for *Escherichia coli*, *Saccharomyces cerevisiae*, and *Chlamydomonas reinhardtii*. However, the flux predictions resulting from enzyme-constrained models are obtained under the implicit assumption that each enzyme is saturated to a constant level, which may be enzyme specific but is condition invariant [21]. These assumptions may not hold for all enzymes and across different experimental scenario considered.

Disentangling the effects of metabolite concentration on a reaction flux entails considering metabolites that act not only as substrates, but also as effectors (*i.e.*, activators and inhibitors). The effect of metabolite concentrations on reaction fluxes captures the magnitude by which metabolites present in the system affect the (condition-dependent) maximal enzyme velocity. Addressing this problem also requires access to absolute and relative quantification of metabolite concentrations in addition to fluxomic and proteomic data [7]. Whilst this strategy can identify metabolites that act as flux effectors, it assumes that all reactions are governed by the same enzyme kinetics. The same assumption is made in other approaches, like model

balancing [8], which estimate enzyme turnover rates while imposing constraints on metabolite concentrations and enzyme abundances. The enzyme kinetics, however, is determined by the underlying reaction mechanism, and these mechanisms differ across reactions included in genome-scale metabolic networks. This raises questions about the validity of the fluxes predicted for unseen experimental conditions and strains by using the fitted enzyme kinetics using the existing strategies.

Here, to address this problem, we propose KineFlux, a hybrid approach rooted in a combination of machine learning and constraint-based modeling. KineFlux predicts genome-scale fluxes that are compatible with metabolite concentration effects given only proteomics data as input. As a result, KineFlux forgoes the specification of the mathematical form of enzyme kinetics, does not require metabolomics data, and capitalizes on the advances in enzyme-constrained metabolic modeling. This is feasible due to the usage of machine learning models for the joint effects of metabolite concentrations on fluxes, using flux-based proxies for metabolite concentrations. By employing data sets from *E. coli* and *S. cerevisiae*, we demonstrated that KineFlux results in excellent prediction of fluxes even for unseen experimental conditions for which only quantitative proteomics data are available.

## Results

### KineFlux is a hybrid of constraint-based and machine learning models

Here, we introduce KineFlux, a hybrid approach that integrates machine-learning-based description of metabolite concentration effects on a reaction flux with constraint-based metabolic modeling. KineFlux consists of two steps: (*i*) determining a function that predicts the metabolite concentration effects on the flux of a given reaction based on metabolite flux-sums, serving as proxies for metabolite concentrations [22], and (*ii*) integrating the resulting functions into a constraint-based modeling approach to predict a flux distribution that is compatible with metabolite concentration effects.

The first step of KineFlux relies on fluxomic and proteomic data to estimate values for the metabolite concentration effects on fluxes across different experimental scenarios (*e.g.*, strains, conditions). Fluxomic data can be obtained by applying parsimonious flux balance analysis (pFBA) using a genome-scale metabolic model (see Methods, Fig 1A) with bounds on the exchange fluxes and growth only [1,7] and/or bounds on intercellular fluxes corresponding to confidence intervals determined by labeling experiments [2]. Like in well-established approaches [6], we used the resulting fluxes to compute the apparent catalytic rate $k_{app}$ for each reaction by calculating the ratio between the estimated flux and the corresponding enzyme abundance obtained from proteomic data (Fig 1B). We then determined the maximum $k_{app}$ value across all scenarios, termed $k_{app}^{max}$, which serves as a proxy for the turnover number [1,2] (Fig 1B). This value was in turn employed to calculate the metabolite concentration effects, $\eta$, for each reaction flux in each strain (Fig 1B). By definition, the metabolite concentration effects on fluxes are in the range [0, 1], that quantifies the reduction in an enzyme's catalytic rate relative to its maximal turnover number. This reduction arises from factors such as incomplete substrate saturation, thermodynamic constraints that promote backward flux, and other *in vivo* environmental limitations.

In the absence of data on metabolite concentration, KineFlux makes use of metabolite flux-sums as proxies for concentrations; this is justified given that steady-state metabolite concentrations can be cast as non-linear (implicit) functions of reaction fluxes. The metabolite concentration effects on a reaction flux, captured by the function $\eta$, considers the reaction substrates as well as (few additional) metabolites that act as regulators (*e.g.*, activators or inhibitors). To this end, the machine learning model for $\eta$ of a reaction in KineFlux relies only on the flux-sums of substrates and at most three additional metabolites in building the machine-learning models. Specifically, we trained and evaluated logit regression models using the flux-sums of reaction substrates along with single, pair, or triplet of other metabolites, employing five-fold cross-validation and selecting the best-performing model based on the adjusted coefficient of determination $R^2$ (Fig 1C and 1D). Using the best performing model, the metabolite concentration effects, $\eta$, can then be predicted based on the flux-sums of metabolites, which can be obtained by linear transformations of fluxes (Fig 1E). We opted to fit logit models to ensure that the predictions based on $\eta$ are in the range [0, 1], in line with biochemical constraints.

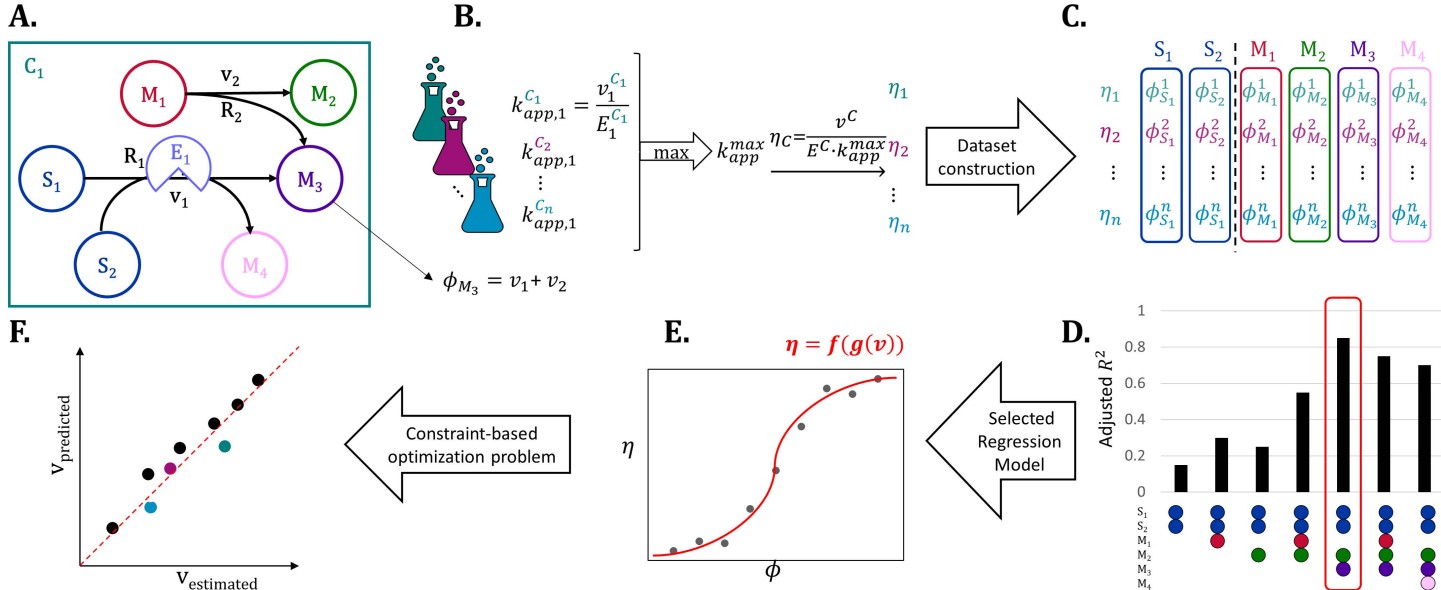

**Fig 1. Schematic overview of KineFlux. A.** A toy example of a metabolic network with two reactions, $R_1$ and $R_2$. Reaction $R_1$ has two metabolites, $S_1$ and $S_2$, acting as substrates, converted into two products, $M_3$ and $M_4$; it is catalyzed by an enzyme with an abundance of $E_1$ and a flux of $v_1$. The flux-sum of $M_3$, $\phi_{M_3}$ is defined as the total flux producing this metabolite. **B.** The apparent turnover number of $R_1$ is determined by dividing the reaction flux by the corresponding enzyme abundance for each experimental scenario (*e.g.*, strain, condition), denoted by a flask of different color. The maximum apparent catalytic rate across all strains $k_{app}^{max}$, is used to calculate the metabolite concentration effects, $\eta$, of the reaction flux for each experiment and reaction with available data. **C.** The features used to predict the metabolite concentration effects, $\eta$, for reaction $R_1$ are the metabolite flux-sums. **D.** Model selection identifies the optimal combination of flux-sums for a single, pair, or a triplet of metabolites to be used as additional predictors together with the flux-sums of the reaction substrates. For reaction $R_1$ in the toy example, the two substrates and four additional metabolites yield C(4,0)+C(4,1)+C(4,2)+C(4,3)=15 combinations, of which seven are displayed. The best model is chosen based on the adjusted $R^2$, identifying the flux-sum of substrates alongside the pair $(M_2, M_3)$. **E.** A logit regression model is then trained to predict $\eta$ as a function of metabolite flux-sums. **F.** Finally, a constraint-based optimization problem, that includes the logit regression models as constraints, is solved to predict a flux distribution and evaluate it against estimated fluxes.

Once the regression models for the metabolite concentration effects over (a subset of) reactions are built and selected, KineFlux integrates them into a constraint-based optimization problem (see Eq (7)). The objective function of this problem includes a quadratic term that minimizes the difference between the predicted flux and the flux formulation in terms of $k_{app}^{max}$, the given enzyme abundance $E$, and $\eta$, predicted by the machine learning models used as constraints. The optimization problem also includes standard steady-state and thermodynamic constraints, ensuring that the predicted flux distribution is in line with biochemical constraints. As a result, KineFlux predicts a flux distribution that is compatible with metabolite concentration effects and can be validated against estimated fluxes (Fig 1F).

## KineFlux results in flux distributions compatible with metabolite concentration effects in *E. coli*

Here, we evaluated the performance of the machine learning models and the flux distributions resulting from the proposed constraint-based optimization problem. We applied our framework to the iJO1366 GEM of *E. coli* [23] using fluxomic and proteomic data from 17 knockout strains each with two biological replicates [2,24–27] (see Methods). To ensure reliable cross-validation, we only retained reactions for which flux measurements were available in more than 10 samples. This filtering step resulted in 514 reactions being retained. Given the limitations of the available protein abundance data, $k_{app}^{max}$ values could be obtained for 562 reactions.

Consequently, we trained logit regression models for 339 reactions in a cross-validation setting and evaluated their performance using the adjusted coefficient of determination ($R^2$). Since two biological replicates were available for each

strain, we grouped the corresponding replicates before performing the data split for cross validation. This ensured that replicates from the same strain were assigned to either the training or the test set together, thereby preventing data leakage. The average and median $R^2$ values were 0.22 and 0.29, respectively, suggesting that while the regression models exhibited good performance ($R^2 > 0.5$) for 92 reactions, there were few reactions for which the selected models were of poor quality. This is further supported by the Fisher-Pearson coefficient of skewness (-5.47), indicating a strong left skewness driven by very few models of poor $R^2$ values (see Fig 2A). These results demonstrated that flux-sums can indeed serve as a reliable feature to predict the metabolite concentration effects, $\eta$, for majority of the reactions for which data were available.

Next, we evaluated the predicted flux distributions for 34 knockout strains by solving the proposed constraint-based optimization problem (see Methods). We assessed the performance of KineFlux by calculating the Pearson correlation between the predicted flux values and the estimated flux for each reaction (see Fig 2B for an example, see S1 Fig and S3 Table for all of the strains). By applying this procedure to all reactions with nonzero flux values present in at least 80% of the knockout strains, we found that the mean and standard deviation of the correlations were 0.86 and 0.27, respectively. In addition, 413 out of 514 reactions for which fluxes were available exhibited a Pearson correlation greater than 0.80 (Fig 2C). These findings indicated that the constraint-based optimization problem, integrating the machine learning regression models, could accurately predict flux distributions consistent with metabolite concentration effects.

To further evaluate the performance of KineFlux, we compared its predicted fluxes with those obtained from established constraint-based approaches, including Flux Balance Analysis (FBA) [28], pFBA, and FBA fluxes predicted from an enzyme-constrained genome-scale model of *E. coli*, named iJO1366* [29]. The flux estimates derived from ${}^{13}$C MFA were used as the ground truth values. For each method, we compared the log-transformed flux distributions against the estimated fluxes and quantified agreement using the Pearson correlation coefficient and the mean squared error (MSE) across all knockout strains. KineFlux consistently outperformed the other flux prediction approaches, exhibiting the highest average Pearson correlation and the lowest average MSE across all tested conditions (Table 1). This finding was supported by a t-test that compared the performance of KineFlux with the second-best method (pFBA) rejected the null hypothesis of equal performance. KineFlux achieved significantly higher Pearson correlations ($p-\text{value} = 3.77 \times 10^{-21}$) and significantly lower MSE values ($p-\text{value} = 5.07 \times 10^{-22}$), confirming that the proposed approach provides more accurate flux predictions across all knockout strains.

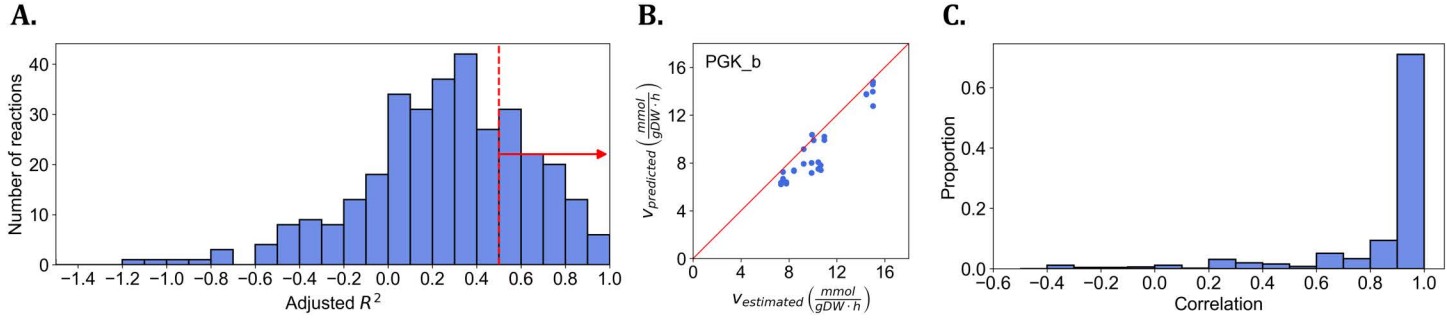

**Fig 2. Performance of logit regression models for metabolite concentration effects and their implication on flux predictions in *E. coli*. A.** The histogram illustrates the performance of the logit regression models in predicting metabolite concentration effects, based on their adjusted $R^2$, for 339 reactions, each with more than 10 values corresponding to different *E. coli* knock-out strains. Among these, 92 reactions achieved an adjusted $R^2$ greater than 0.5. The respective logit models were in turn used in the constraint-based optimization problem **B.** Comparison of the predicted flux from the optimization problem with the estimated flux for the phosphoglycerate kinase (PGK_b) reaction, resulting in a Pearson correlation coefficient of 0.90 (p-value = $2.11 \times 10^{-13}$). **C.** The histogram presents the number of reactions based on the Pearson correlations between their predicted and estimated fluxes.

**Table 1. Comparison of KineFlux performance with established constraint-based modeling approaches.**

| | log-transformed flux distributions | | flux of the reactions | |
|---|---|---|---|---|
| | Pearson | MSE | $R^2$ | MSE |
| KineFlux | $0.859 \pm 0.043$ | $4.464 \pm 1.530$ | $0.866 \pm 0.259$ | $0.388 \pm 1.744$ |
| FBA | $0.450 \pm 0.060$ | $28.819 \pm 5.473$ | $0.574 \pm 0.508$ | $1.484 \times 10^5 \pm 2.83 \times 10^5$ |
| pFBA | $0.692 \pm 0.036$ | $9.372 \pm 1.539$ | $0.803 \pm 0.367$ | $0.677 \pm 3.213$ |
| FBA on ecGEM | $0.619 \pm 0.026$ | $13.599 \pm 1.068$ | $0.463 \pm 0.480$ | $2.56 \times 10^4 \pm 1.204 \times 10^5$ |

Shown is the comparison of fluxes predicted from KineFlux with those from established constraint-based approaches, including FBA, pFBA, and an FBA fluxes obtained from ecGEM. The genome-scale metabolic model iJO1366 of *E. coli* was used, and flux estimates derived from ¹³C MFA were considered as the ground truth. The performance was evaluated using the correlation coefficient and MSE as metrics. The second and third columns present the mean and standard deviation of these metrics considering strain-specific log-transformed flux distributions across all strains. The fourth and fifth columns report the corresponding mean and standard deviation of the metrics computed per reaction across all strains.

Moreover, we evaluated the coefficient of determination ($R^2$) and mean squared MSE of flux predictions from KineFlux and other mentioned constraint-based approaches, considering only reactions with at least 80% non-zero estimated flux values across all knockout strains. KineFlux outperformed the other methods, exhibiting higher $R^2$ values and lower MSE (Table 1). To assess the statistical significance of these improvements, we performed paired t-tests comparing KineFlux with the second-best method, pFBA. KineFlux achieved significantly higher $R^2$ ($p-value = 9.39 \times 10 \times 10^{-6}$) and lower MSE ($p-value = 9.33 \times 10^{-3}$), demonstrating that the proposed approach provides more accurate flux predictions compared to other constraint-based methods.

Furthermore, we analyzed the distribution of $\eta$ values obtained from the constraint-based optimization in two ways. First, when considering each strain individually, the mean and standard deviation of $\eta$ across all strains were 0.46 and 0.08, respectively. Second, when examining the distribution of $\eta$ for each reaction across the strains. The mean of $\eta$ values for the reactions ranged from $0.06 \pm 0.05$ for FAD reductase to $0.77 \pm 0.33$ for Fatty-acid-CoA thioesterase. Overall, the mean and standard deviation for all reactions included in the optimization were 0.455 and 0.175, respectively. For comparison, the $\eta$ values of the reactions in the training data, for the same set of reactions, exhibited a mean and standard deviation of 0.41 and 0.16, respectively. These findings indicate that KineFlux produces moderately variable $\eta$ values.

### Reactions with well-predicted fluxes compatible with metabolite concentration effects are enriched in key metabolic systems

Next, we evaluated the quality of the predicted fluxes for each strain and subsystem in the utilized GEM. To this end, we compared the log-transformed estimated and predicted fluxes using the deviation from the best regression line. We considered fluxes to be well-predicted if they were in the prediction interval band, corresponding to a 90% confidence level for the log-transformed predicted and estimated fluxes (Fig 3A). The mean and standard deviation of the percentage of well-predicted fluxes across all strains were 0.981 and 0.004, respectively, indicating excellent performance of KineFlux.

We also determined the percentage and standard deviation for the number of well-predicted fluxes in each metabolic subsystem and tested for enrichment (Fig 3B). We identified 14 metabolic subsystems to be enriched with well-predicted across all strains. These subsystems contribute to biomass and maintenance functions, tRNA charging, cell envelope biosynthesis and structural integrity, amino acid metabolism, and nitrogen metabolism. Therefore, we concluded that most reactions in central metabolism are of well-predicted fluxes, demonstrating that the KineFlux approach effectively integrates metabolite concentration effects for these pathways.

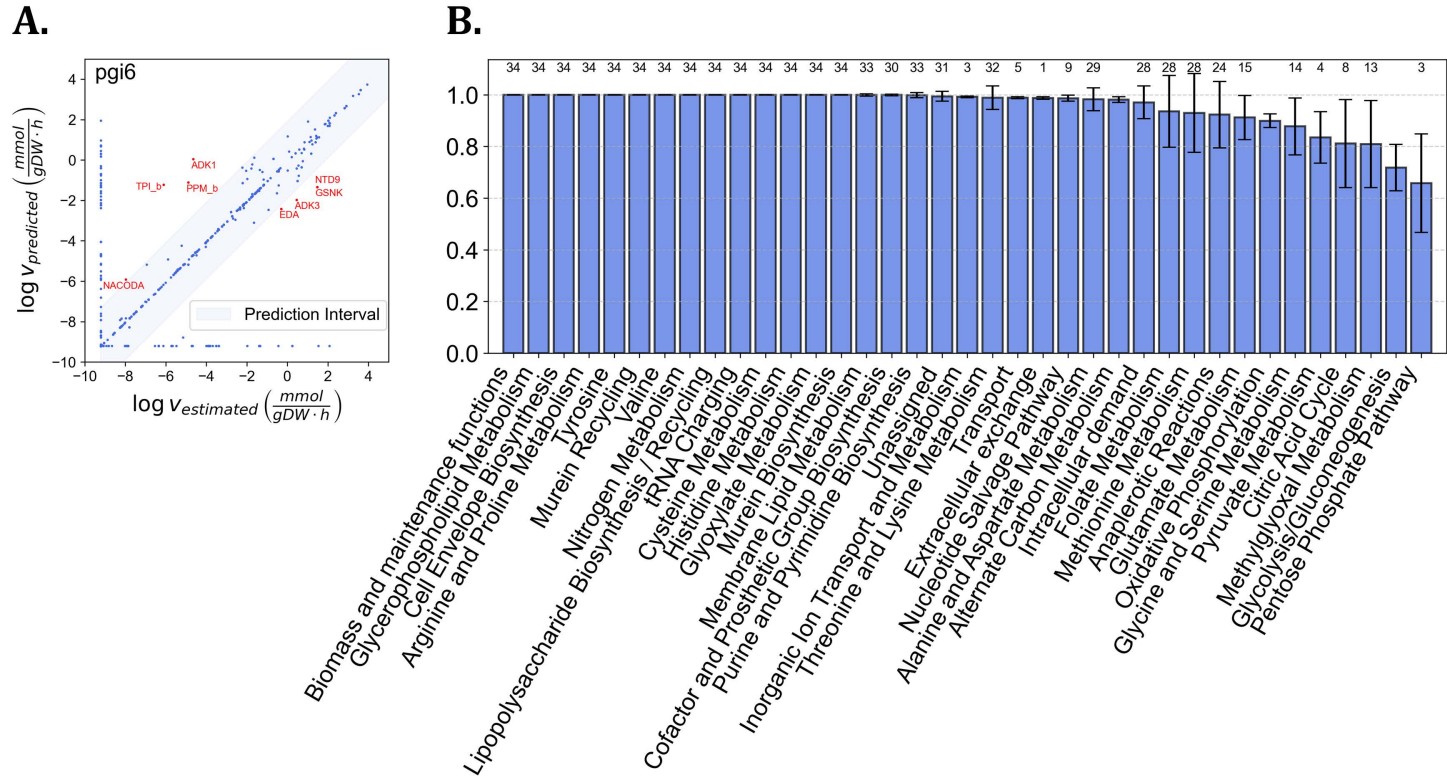

**Fig 3. Enrichment analysis for reactions with well-predicted fluxes. A.** Comparison of a predicted and estimated flux distribution for a representative knock-out strain, *pgi6*. The fluxes are logarithmically transformed, with a small constant ($10^{-4}$) added to all values to avoid taking the logarithm of zero. The Pearson correlation between the predicted and estimated fluxes is 0.87 (p-value = 0.0). A prediction interval band, corresponding to a 90% confidence level, is depicted in light blue. The reactions inside this interval band are considered to have well-predicted fluxes. Highlighted reactions outside of the confidence region include: TPI_b (Triose-phosphate isomerase), ADK1 (Adenylate kinase), PPM_b (Phosphopentomutase), GSNK (Guanosine kinase), NTD9 (5'-nucleotidase (GMP)), ADK3 (Adentylate kinase (GTP)), NACODA (N-acetylornithine deacetylase), and EDA (2-dehydro-3-deoxy-phosphogluconate aldolase) **B.** The mean and standard deviation of the proportions of reactions with well-predicted fluxes across subsystems for all knock-out strains. The value above each bar indicates the number of knockout strains in which the corresponding metabolic subsystem is significantly enriched with reactions exhibiting well-predicted fluxes, determined using a hypergeometric test with Bonferroni-corrected p-values below the 0.02 significance threshold.

## KineFlux points at enzyme regulators

In this section, we investigated whether the effectors identified by the logistic models correspond to known regulators of enzyme activity. To this end, for each selected reaction, we incorporated the three top metabolites identified as effectors in the regression models, along with alternative metabolites exhibiting a strong feature correlation (Pearson correlation of coefficients greater than 0.8) with those of the considered three metabolites. Subsequently, we compared the found sets of putative regulators with known effectors of enzyme activities available in BRENDA [30] and STITCH [31] databases for *E. coli*.

Using the BRENDA database, we retrieved 123 regulatory interactions associated with enzymes catalyzing 60 reactions annotated with Enzyme Commission (EC) numbers, for which machine learning models were trained and demonstrated satisfactory performance (see S1 and S2 Tables). Among these regulators, five were also identified in our approach. For example, two effectors of homoserine kinase, as well as one effector each for sedoheptulose-1, 7-bisphosphate D-glyceraldehyde-3-phosphate-lyase, porphobilinogen synthase, and serine O-acetyltransferase, identified by KineFlux, were likewise reported as effectors in BRENDA.

Using data from the STITCH database, we achieved an overall precision of 11.96% and recall of 8.13% across 92 reactions. When evaluated on a per-reaction basis, the precision values ranged from 0% to 60%, with mean and median values of 15.5% and 9.83%, respectively. Notably, at least one known interaction in STITCH was identified for 75 reactions. This suggests possibility of refining the proposed hybrid approach in the direction of detection of metabolite-enzyme interactions, that has been attempted using features engineered from genome-scale metabolic models [32,33]. We note that more extensive testing and usage of other machine learning approaches may be required to address to systematically address this question.

## KineFlux is applicable to metabolic networks of eukaryotes without changes in performance

Next, we assessed the performance of KineFlux for the eukaryotic model organism, *S. cerevisiae*, utilizing the Yeast-GEM [34] and incorporating fluxomic and proteomic data from 62 different growth conditions [35–39]. We constructed a dataset for 223 reactions (4% of the number of reactions in the model) that, as in the case of *E. coli*, are catalyzed by single enzymes with available abundance data, had non-zero fluxes in at least 10 samples, as well as values for for the metabolite concentration effects in the range of (0, 1]. The average and median adjusted $R^2$ values of the logit models in this case were 0.42 and 0.46, respectively, suggesting that while the regression model performed well ($R^2 > 0.6$) for 73 reactions, like in the case of *E. coli*, there were a few reactions for which the selected models were of poor quality. Here, too, we found left-skewed distribution of adjusted $R^2$ values (Fisher-Pearson coefficient of skewness of -2.09) (see Fig 4A). These results indicated that metabolite flux-sums can serve as a reliable feature to predict metabolite concentration effects regardless of the organism.

Next, we evaluated the predicted flux distributions across the 62 considered conditions (see Methods). To this end, we calculated the Pearson correlation between the predicted flux values and the estimated flux for each reaction (see Fig 4B as an example, see S2 Fig for all conditions). Applying this procedure to all reactions with more than 80% nonzero fluxes, we found that the mean and standard deviation of the correlations were 0.82 and 0.29, respectively, and that 299 out of 418 reactions (71.5%) exhibited a Pearson correlation greater than 0.8 (see Fig 4C). Moreover, we found that the mean and standard deviation of the Pearson correlation between the log-transformed predicted and estimated fluxes (for an example, see Fig 4D) in each condition were 0.81 and 0.05, respectively. Like in the case of *E. coli*, these results demonstrated that KineFlux can be effectively used to predict accurate flux distributions compatible with metabolite concentration effects across conditions in *S. cerevisiae*.

Finally, we assessed the quality of the predicted fluxes for each condition and each subsystem of the utilized yeast genome-scale metabolic model, using the same definition of well-predicted fluxes as in *E. coli*. The mean and standard deviation of the percentage of well-predicted fluxes across all conditions were 0.99 and 0.004, respectively (Fig 4D). We identified 10 subsystems containing more than 30 reactions, all of which were well-predicted. Among these, seven are a part of lipid metabolism, as one of the central metabolic processes (Fig 4E).

## KineFlux is transferrable to unseen condition with little effect on accuracy

To further evaluate KineFlux, we assessed its ability to predict flux distributions under previously unseen conditions. To this end, we employed the logit models trained on the data set from Heckmann *et al.* [2] (see Methods) and integrated them into the constraint-based optimization problem to predict flux distributions for the testing conditions used in Davidi *et al.* [1], corresponding to growth on different carbon sources.

We considered two scenarios regarding the $k_{app}^{max}$ values and protein abundances used. In the first scenario, we used the enzyme abundances from the unseen condition with $k_{app}^{max}$ values derived from the training data. By solving the problem for 26 testing conditions, we found that the mean and standard deviation of the Pearson correlation coefficients between predicted and estimated fluxes of the reactions across conditions were 0.69 and 0.37, respectively. Notably, 69% of the reactions exhibited correlations greater than 0.5 (see Fig 5A). In contrast, the mean correlation when applying the same

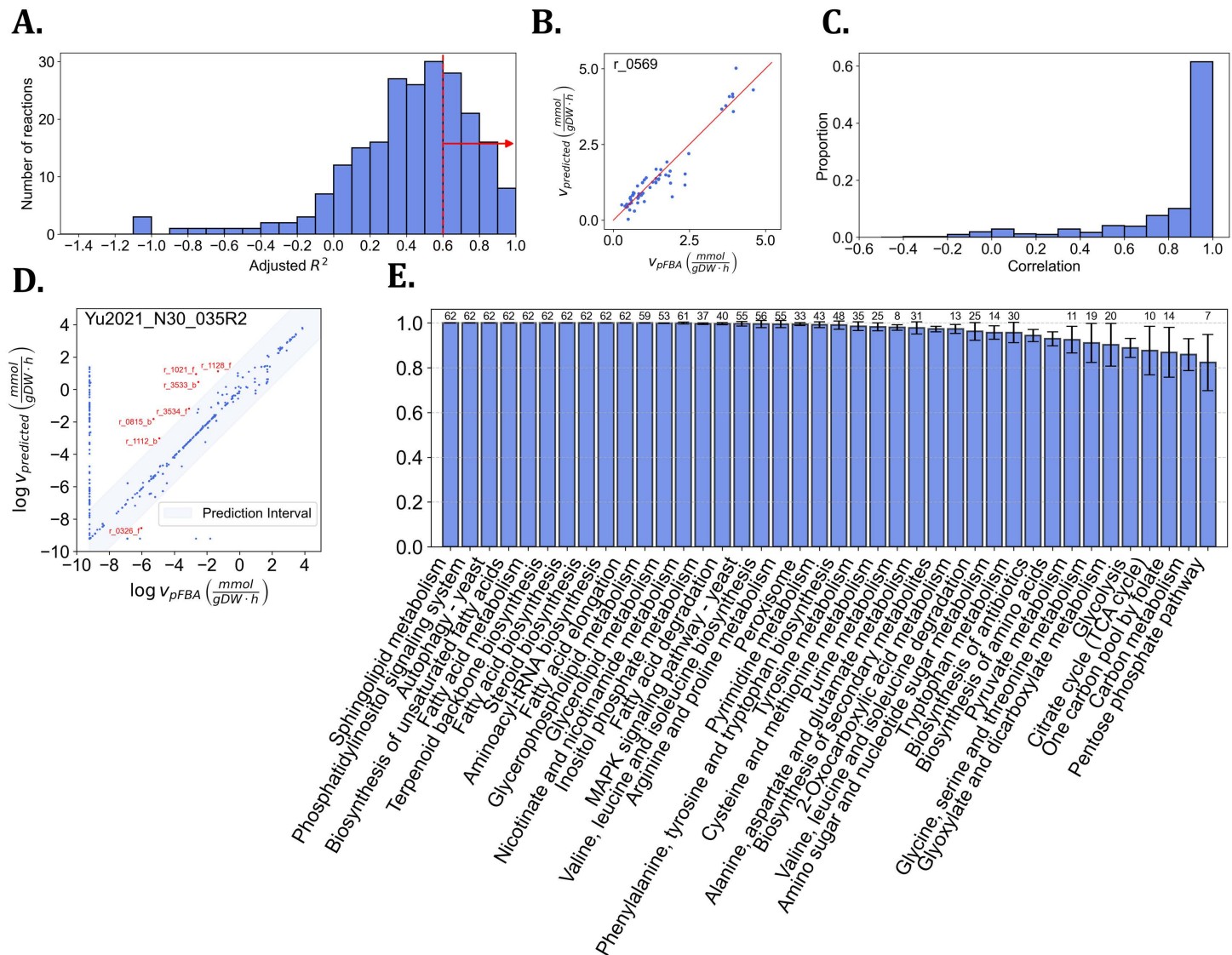

**Fig 4. Performance of logit regression models for metabolite concentration effects and evaluation of predicted flux distributions in *S. cerevisiae*. A.** The histogram illustrates the performance of the logit regression models in predicting metabolite concentration effects, $\eta$, based on their adjusted $R^2$. The data set comprises 281 reactions, each with more than 10 entries corresponding to different *S. cerevisiae* conditions. Among these, 73 reactions achieved an adjusted $R^2$ greater than 0.6, used in the constraint-based optimization problem. **B.** The plot compares the predicted flux from the optimization problem with the estimated flux for the reaction r_0569 (inorganic diphosphatase), resulting in a Pearson correlation coefficient of 0.95 (p-value= $4.89 \times 10^{-33}$. **C.** The histogram presents the number of reactions based on the Pearson correlations between their predicted and estimated fluxes. In total, there are 418 reactions with at least 80% non-zero estimated fluxes across different conditions. More than 80% of these reactions have a Pearson correlation greater than 0.8 between estimated and predicted fluxes. **D.** Comparison of the predicted flux distribution with the estimated flux distribution for a representative condition, Yu2021_N30_035R2, which corresponds to the second biological replicate of nitrogen-limited chemostat growth at a dilution rate of 0.35 $h^{-1}$ and a carbon-to-nitrogen (C/N) ratio of 30 [39]. The fluxes are logarithmically transformed, with a small constant ($10^{-4}$) added to all values to avoid taking the logarithm of zero. The Pearson correlation between the predicted and estimated fluxes is 0.86 (p-value = 0.0). A prediction interval band, corresponding to a 90% confidence level, is included. The reactions inside the prediction interval band are considered to have well-predicted fluxes. Highlighted reactions outside of the confidence region include: r_1021_f (succinate dehydrogenase (ubiquinone-6)), r_0815_b (O-succinylhomoserine lyase (L-cysteine)), r_0326_f (dCMP deaminase), r_3533_b (NAD transport, cytoplasm-ER membrane), r_1128_f (citrate transport), r_3534_f (glycerol 3-phosphate transport, cytoplasm-ER membrane), and r_1112_b (AKG transporter) **E.** The mean and standard deviation of the proportions of reactions with well-predicted fluxes across subsystems for all conditions. We limited the subsystem to those with more than 30 reaction. The value above each bar indicates the number of conditions in which the subsystem is significantly enriched with reactions exhibiting well-predicted fluxes, determined using a hypergeometric test with Bonferroni-corrected p-values below the 0.02 threshold.

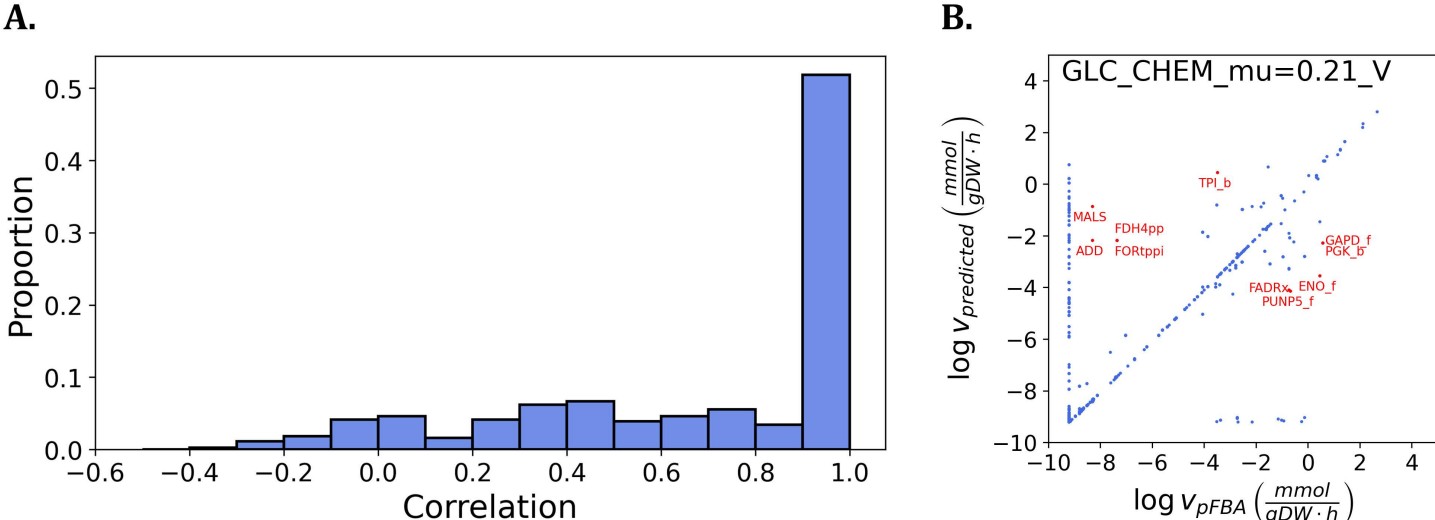

**Fig 5. Performance of KineFlux to predict flux distribution for unseen conditions in *E. coli*. A.** Histogram showing the distribution of Pearson correlation coefficients between predicted and estimated fluxes of the reactions across various growth conditions, each associated with different carbon uptake sources [1]. **B.** Comparison of the predicted and estimated fluxes under the condition GLC_CHEM_mu=0.21_V from Davidi *et al.* [1], which corresponds to a chemostat culture with a growth rate of 0.21 $h^{-1}$ using glucose as the carbon source [40]. All flux values are log-transformed, with a small constant ($10^{-4}$ added to prevent logarithms of zero. Highlighted reactions located farthest from the diagonal, with non-zero predicted fluxes, include: MALS (Malate synthase), ADD (Adenine deaminase), FORtppi (Formate transport via diffusion), FDH4pp (Formate dehydrogenase (quinone-8)), ENO_f (Enolase), TPI_b (Triose-phosphate isomerase), PUNP5_f (Purine-nucleoside phosphorylase (Inosine)), FADRx (FAD reductase), GAPD_f (Glyceraldehyde-3-phosphate dehydrogenase), and PGK_b (Phosphoglycerate kinase).

procedure to the training data was 0.85 (see above), indicating a performance drop when generalizing to new conditions. This reduction in predictive accuracy can likely be attributed to two main factors. First, the regression models were trained on only 17 knock-out strains, that may limit their generalization capacity. In support of this claim, 22 out of 92 reactions with available regression models exhibited a correlation below 0.5 (out of 133 total reactions). Second, the discrepancy may arise from the differing sources of flux estimates: the flux estimates in the unseen condition are based on pFBA, whereas the training data relies on more accurate $^{13}$C MFA. Despite these limitations, our results demonstrate that KineFlux can effectively predict flux distributions in previously unseen conditions using only enzyme abundance data, with only a modest decline in accuracy.

In the second scenario, we incorporated not only enzyme abundances from the unseen conditions, but also the corresponding $k_{app}^{max}$ values into the constraint-based optimization problem (see Eq (7)). In this scenario, the machine learning models for the metabolite concentration effects were applied only to those reactions for which logit models were available from Heckmann *et al.* and for which both $k_{app}^{max}$ and enzyme abundance data were available from the unseen, testing conditions of Davidi *et al.*. As a result, we found that the mean and standard deviation of the Pearson correlation coefficients between the predicted and estimated fluxes across reactions were 0.781 and 0.347, respectively.

To enable a fair comparison between the two aforementioned scenarios, we repeated the flux predictions under the criteria for the first scenario and restricting the machine learning models to exactly the same subset of reactions. In this case, the mean and standard deviation of the correlations were 0.853 and 0.297, respectively. A paired t-test showed that the performance of the first scenario significantly greater than that of the second scenario ($p-value = 8.55 \times 10^{-4}$), indicating that $k_{app}^{max}$ values derived from knockout strains (Heckmann et al. [2]) resulted in better flux prediction than those obtained under different growth conditions (Davidi et. al. [1]).

## Discussion

Despite advances in enzyme-constrained genome-scale metabolic models, constraint-based modeling approaches are inherently flux-centric. Although flux distributions predicted from enzyme-constrained metabolic models can reproduce metabolic phenotypes, like the Crabtree effect [18], we lack understanding of the effects that metabolite concentrations have on fluxes under different experimental scenarios. This challenge persists since matched quantitative fluxomic, proteomic, and metabolomic data sets for different strains or conditions are still difficult to assemble. Our aim was to expand the applicability of enzyme-constrained models to predict steady-state distributions compatible with metabolite concentration effects by relying solely on proteomic data. This is particularly useful to predict flux distributions in scenarios where labeling experiments necessitate substantial resources and efforts even in a single cell organism, like *E. coli* (*e.g.,* the study of nitrogen metabolism using instationary labeling studies).

Here, we devised, implemented, and tested KineFlux, a hybrid constraint-based modeling approach that relies on constraints from machine learning models for the metabolite concentration effects on fluxes. These machine learning models are trained using metabolic concentration effects derived from matched fluxomic and proteomic data. In training the models, we ensured that the predicted metabolite concentration effects are bounded in the interval (0, 1], in line with biochemical constraints, contributing to their transferability to unseen conditions. Indeed, our results demonstrate that the flux distributions predicted by KineFlux are not reliable only for seen conditions, but are transferrable without significant loss in accuracy to unseen conditions (even with altered, but concordant estimates for the turnover numbers), as we showed for *E. coli*.

In absence of metabolomic data, we trained models for the metabolite concentration effects in terms of transformation of fluxes, in the form of metabolite flux-sums. The usage of metabolite flux-sums facilitates the interpretability of the machine learning models and guides their training, achieved using these features for substrates and any combination of at most three effectors for each reaction flux. Based on first principles, one would expect that higher substrate availability would have positive effects on reaction flux; however, from the 259 substrates included in the machine learning models for the metabolite concentration effects of 113 reactions in *E. coli*, 146 have positive coefficients and 113 negative coefficients in the logit models. The presence of negative coefficients is partly in line with substrate inhibition, as the most common deviation from Michaelis-Menten enzyme kinetics, occurring in over 25% of known enzymes [41,42]. However, further extensions and refinements to KineFlux are needed to increase the interpretability of the metabolite-enzyme interactions.

In summary, KineFlux provides an innovative usage of enzyme-constrained metabolic models and expands their predictive capacity. Further, it opens the possibility for usage of other machine learning models as constraints in large-scale metabolic modeling, allowing metabolite concentration effects to be considered even for enzymes whose enzyme kinetics is challenging to pinpoint. Finally, KineFlux allows us to bridge the gap between concentration- and flux-centric analyses that currently separate kinetic and constraint-based metabolic modeling.

## Methods

The main aim of the proposed approach, termed KineFlux, is to predict flux distributions compatible with metabolite concentration effects. To this end, first we present definitions used in flux modeling necessary for the creation of a data set comprising flux-sums and the metabolite concentration effects on a flux, denoted by $\eta$. Second, we explain the machine learning (ML) regression model to predict $\eta$ based on the flux-sums. Finally, we used the results from the preceding steps to formulate the constraint-based optimization problem to predict a flux distribution compatible with metabolite concentration effects.

### Preprocessing and definitions

We used fluxomics and proteomic data and several definitions to construct the data set for predicting metabolic concentration effects on fluxes. We used the confidence intervals for reaction fluxes from ${}^{13}$C metabolic flux analysis (MFA) [2], obtained using the genome-scale metabolic network model of *E. coli* iJO1366 [23]. This is the reason for retaining the use

of the same model for the rest of the analyses, since change of model can have effect on the flux estimates [43]. The data set includes 17 strains each with two biological replicates based on gene knockout of phosphotransferase system (*pts*) [25], phosphoglucose isomerase (*pgi*) [27], triosephosphate isomerase (*tpi*) [24], and succinate dehydrogenase (*sdh*) [26] with four, seven, three, and three strains, respectively. Since flux confidence intervals were available for fewer than 12.5% of the reactions in the model, we solved the parsimonious flux balance analysis (pFBA) [14] with the irreversible iJO1366 model while respecting the constraints on the lower and upper bounds of the fluxes from measurements to obtain a flux distribution for the entire model. We used the obtained fluxes to: (*i*) determine of the enzyme kinetic parameters, including $k_{app}^{max}$ and $\eta$; (*ii*) calculate flux-sum of the metabolites.

First, we note that the flux, $v$, through a reaction catalyzed by an enzyme is determined by the turnover number, $k_{cat}$, of the enzyme, the enzyme abundance, $E$, and the metabolite concentration effects, captured by the function $\eta$, with a range in the interval $[0, 1]$. This function describes the effect of concentration of metabolites (*e.g.*, substrates, activators, and inhibitors) on fluxes. This can be mathematically written in the following form:

$$v = k_{cat} \cdot E \cdot \eta. \tag{1}$$

To address the low-throughput measurements of $k_{cat}$ values, that are still challenging to obtain on a systems level, and the expected discrepancies between *in vitro* and *in vivo* $k_{cat}$ values, one can determine condition-specific apparent catalytic rate ($k_{app}^{C}$) by employing flux estimates and protein abundances measurements using the following equation [1,2]:

$$k_{app}^{C} = \frac{v^{C}}{E^{C}}. \tag{2}$$

The $k_{app}^{max}$, defined as the maximum of $k_{app}^{C}$ over all considered conditions (or strains) $C$ can serve as a proxy for $k_{cat}$ [1,2]. Therefore, by considering Eq (1), we can calculate condition-specific $\eta^{C}$ as follows:

$$\eta^{C} = \frac{v^{C}}{E^{C} \cdot k_{app}^{max}}. \tag{3}$$

Second, we define the flux-sum of a metabolite as the sum of all fluxes that contribute to the production of the metabolite, weighted by their molecular coefficients, *i.e.*,

$$\phi_i = \sum_{\forall j \in R} S_{ij}^{+} v_j, \tag{4}$$

where $S_{ij}^{+}$ represents the stoichiometric coefficient of metabolite $i$, acting as a product of reaction $j$.

## Prediction of metabolite concentration effects on fluxes

It has been shown that flux-sums can serve as a proxy for metabolite concentrations [22]. With this in mind, and having calculated the flux-sums for all metabolites and the values for reactions, we trained a logit regression model (see Eq 6) to predict a value for the metabolite concentration effects using metabolite flux-sums. For each reaction, the response variable consisted of the values of $\eta$ across different strains or conditions, while the predictor variables included the flux-sums of the reaction's substrates. In addition, we incorporated flux-sums of other metabolites, selected based on their potential to enhance model performance. Specifically, we considered not only the flux-sums of the substrates but also those of single metabolites, as well as pairs and triplets of other metabolites. The best-performing model was selected from the different variants based on these predictor combinations, using five-fold cross-validation with the adjusted coefficient of determination

($R^2$) as the performance metric. For each fold, we trained the regressor and computed the average metric across the folds. Ultimately, we identified the metabolite combination for each reaction that maximized the adjusted $R^2$ value.

We decided to use logit regression as it ensures that the predicted values remain within the range [0,1], providing a constrained and interpretable output:

$$\eta = \frac{1}{1 + e^{-f(\phi)}},$$
$$f(\phi) = \beta + \sum_{k \in K} \alpha_k \phi_k, \tag{5}$$

where $\alpha_k$ denotes a regression coefficient, $\beta$ is the intercept, and $K$ is a subset of selected metabolites whose flux-sums are used as predictors of $\eta$. Note that $\varphi$ depends on the flux values of the metabolites $\phi = g(v) = S^+v$, which are incorporated as predictors in the model. Therefore, the predicted value ultimately depends on these flux values:

$$\eta = \frac{1}{1 + e^{-f(g(v))}}, \tag{6}$$

## Prediction of flux distributions compatible with metabolite concentration effects

At the last step of the approach, we integrated the logit models for $\eta$ of the reactions, for which data were available for training models of good quality, in an optimization problem whose solution is a flux distribution which is compatible with enzyme kinetic. To this end, for a condition $C$, we solve the following optimization problem:

$$\min \quad \sum_{j \in P} \| v_j - E_j^C \cdot k_{\text{app},j}^{\max} \cdot \eta_j^{\text{pred},C} \|_2 + w \sum_{i \in R} v_i$$

$$\text{s.t.} \quad \eta_j^{\text{pred}} = \frac{1}{1 + e^{-f_j(\phi)}},$$

$$f_j(\phi) = \sum_{k \in K} \alpha_k \phi_k + \beta,$$

$$\Phi = S^+ v,$$

$$S \cdot v = 0,$$

$$v_l \le E_l^C \cdot k_{\text{app},l}^{\max}$$

$$v^{lb} \le v \le v^{ub},$$

$$v \in \mathbb{R}^{|R|}, \quad \eta^{\text{pred}} \in [0, 1]^{|P|}, \quad j \in P, \quad l \in Q, \tag{7}$$

where $\eta^{\text{pred}}$ is a predicted value of $\eta$ for the reaction using learned parameters based on the flux-sums, $P$ is the set of reactions whose regressor model is evaluated as $R^2 > 0.6$ in the five-fold cross-validation setting, $Q$ is the subset of all reaction for which protein abundance and $k_{\text{app}}^{\max}$ is available, and $w$ is a weight for the summation of all fluxes in the objective function. We used a small value ($w = 1e^{-2}$) so that the optimization favors the contribution of the quadratic term in the optimization function that considers the contribution from the metabolite concentration effects on fluxes of multiple reactions. All constraint-based optimization problems were solved using the Gurobi Optimizer [44]

## Supporting information

**S1 Fig. Comparison of predicted and measured flux distributions in *E. coli*.** The predicted flux distribution from the constraint-based optimization problem was compared to the measured flux across seventeen different knock-out strains, each with two biological replicates. The average and standard deviation of the Pearson correlations between predicted and measured fluxes across all strains were 0.849 and 0.0313, respectively.
(TIF)

**S2 Fig. Comparison of predicted and measured flux distributions in *S. cerevisiae*.** The predicted flux distribution from the constraint-based optimization problem was compared to the measured flux under 62 conditions. The average and standard deviation of the Pearson correlations between predicted and measured fluxes across all strains were 0.812 and 0.0504, respectively.
(TIF)

**S1 Table. Regulators retrieved from BRENDA and predicted by logistic regression models.** This table lists reactions with an adjusted $R^2$ greater than 0.6, associated with an EC number in *E. coli*. For each reaction, up to three top candidate regulators identified by logistic regression are included, along with those showing a flux-sum correlation greater than 0.8. The table also includes compound IDs (CIDs) of retrieved regulators from BRENDA and highlights the overlap between these and the model-predicted regulators.
(XLSX)

**S2 Table. Metabolite names corresponding to the compound IDs (CIDs) in S1 Table.** This table provides the names of metabolites associated with the CIDs listed in S1 Table.
(XLSX)

**S3 Table. Evaluation of KineFlux in *E. coli*.** The table reports the performance of KineFlux across seventeen knockout strains, each with two biological replicates, using Pearson correlation and mean squared error (MSE) between measured and predicted fluxes.
(XLSX)

## Author contributions

**Conceptualization:** Zoran Nikoloski.

**Formal analysis:** Fayaz Soleymani, Zahra Razaghi-Moghadam.

**Investigation:** Fayaz Soleymani, Zoran Nikoloski.

**Methodology:** Fayaz Soleymani, Zahra Razaghi-Moghadam, Zoran Nikoloski.

**Project administration:** Zoran Nikoloski.

**Software:** Fayaz Soleymani.

**Supervision:** Zoran Nikoloski.

**Validation:** Zoran Nikoloski.

**Visualization:** Fayaz Soleymani, Zoran Nikoloski.

**Writing – original draft:** Fayaz Soleymani, Zoran Nikoloski.

**Writing – review & editing:** Fayaz Soleymani, Zahra Razaghi-Moghadam, Zoran Nikoloski.

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
