## [Decision Letter · Decision Letter 0]

23 Sep 2025

Accurate prediction of flux distributions compatible with metabolite concentration effects in genome-scale metabolic networks

PLOS Computational Biology

Dear Dr. Nikoloski,

Thank you for submitting your manuscript to PLOS Computational Biology. After careful consideration, we feel that it has merit but does not fully meet PLOS Computational Biology's publication criteria as it currently stands. Therefore, we invite you to submit a revised version of the manuscript that addresses the points raised during the review process.

Please submit your revised manuscript within 60 days Nov 23 2025 11:59PM. If you will need more time than this to complete your revisions, please reply to this message or contact the journal office at ploscompbiol@plos.org. Please include the following items when submitting your revised manuscript:

We look forward to receiving your revised manuscript.

Kind regards,

Vassily Hatzimanikatis

Academic Editor

PLOS Computational Biology

Pedro Mendes

Section Editor

PLOS Computational Biology

**Journal Requirements:**

At this stage, the following Authors/Authors require contributions: Zahra Razaghi-Moghadam, Fayaz Soleymani, and Zoran Nikoloski. Please ensure that the full contributions of each author are acknowledged in the "Add/Edit/Remove Authors" section of our submission form.

5) We have noticed that you have uploaded Supporting Information files, but you have not included a complete list of legends. Please add a full list of legends for your Supporting Information files (Suplementary Tables) after the references list.

6) We notice that your supplementary Figures are included in the manuscript file. Please remove them and upload them with the file type 'Supporting Information'. Please ensure that each Supporting Information file has a legend listed in the manuscript after the references list.

7) Some material included in your submission may be copyrighted. According to PLOSu2019s copyright policy, authors who use figures or other material (e.g., graphics, clipart, maps) from another author or copyright holder must demonstrate or obtain permission to publish this material under the Creative Commons Attribution 4.0 International (CC BY 4.0) License used by PLOS journals. Please closely review the details of PLOSu2019s copyright requirements here: PLOS Licenses and Copyright. If you need to request permissions from a copyright holder, you may use PLOS's Copyright Content Permission form.

Potential Copyright Issues:

i) Figure 1B. Please confirm whether you drew the images / clip-art within the figure panels by hand. If you did not draw the images, please provide (a) a link to the source of the images or icons and their license / terms of use; or (b) written permission from the copyright holder to publish the images or icons under our CC BY 4.0 license. Alternatively, you may replace the images with open source alternatives. See these open source resources you may use to replace images / clip-art:

8) Please amend your detailed Financial Disclosure statement. This is published with the article. It must therefore be completed in full sentences and contain the exact wording you wish to be published.

9) Please ensure that the funders and grant numbers match between the Financial Disclosure field and the Funding Information tab in your submission form. Note that the funders must be provided in the same order in both places as well. Currently, "the NovoNordisk Foundation, grant number NNF23OC0085412" is missing from the Funding Information tab.

**Reviewers' comments:**

Reviewer's Responses to Questions

Reviewer #1: Soleymani et al. introduce KineFlux, a flux balance analysis (FBA) approach that integrates machine learning with enzyme-constrained modeling to incorporate metabolite effects. While the methodology appears elegant, its novelty requires more thorough demonstration. I hope the following comments will help improve the manuscript.

First, I do not believe the primary novelty of KineFlux lies in its ability to predict more accurate fluxes compared to other FBA methods. If the authors intend to claim this as a key advance, they should provide supporting evidence e.g. by demonstrating that KineFlux outperforms state-of-the-art FBA methods in flux predictions.

Furthermore, it remains unclear whether there exists any specific scenario where only KineFlux is applicable, while other methods are not. The authors note that KineFlux can predict flux distributions using only quantitative proteomics data (Lines 15-16). However, classical FBA can also predict fluxes with only a few exchange rates as constraints, which are often more experimentally accessible than omics data. Even enzyme-constrained FBA models can predict fluxes without any experimental input by simply maximizing the growth rate while allowing unlimited nutrient uptake.

In my view, the true novelty of KineFlux may lie in its ability to infer reactions regulated by metabolite concentrations. This aspect should be emphasized and further demonstrated in a revised version. The authors could focus more on the predicted metabolite-mediated regulation e.g. by evaluating how many of these predictions are supported by existing experimental evidence from literature and databases, and explaining why KineFlux might fail to capture some known regulatory effects. Presenting these results in the main figures, rather than only discussing them in the text, would strengthen the argument.

Although the authors mention that KineFlux utilizes enzyme-constrained models, they do not detail how the framework integrates such constraints. The Methods section lacks information on the construction of enzyme-constrained models for E. coli and S. cerevisiae. Additionally, it would be valuable to clarify whether KineFlux can be applied to classical metabolic models without enzyme constraints.

In addition, I have some minor comments below:

Lines 24-25: “without knowledge of enzyme kinetics”, but the enzyme-constrained models already include enzyme kinetics.

Fig 1D: what is the meaning of the adjusted R2 value? Does it correspond to the coefficient for the fitting curve in Fig 1E?

Line 94: double “that”.

Line 235: the authors mentioned Warburg effect but the reference [18] is about S. cerevisiae. Maybe the authors meant Crabtree effect?

Reviewer #2: The manuscript presents a method for predicting the modulation of metabolic fluxed by non-substrate metabolites. For this, the authors introduce a function ‘eta’ that takes values between zero and one and gets multiplied with a predicted flux to account for regulatory effects of other metabolites. The function eta is modelled as a logistic regression estimated from metabolomic and proteomic datasets, with at most three non-substrate metabolites in used as covariates. Overall, the method looks promising and the authors demonstrate the applicability on a number of datasets from E. coli and yeast.

Nevertheless, the manuscript has a number of major shortcomings that I would like to see addressed:

1) Please be clearer and more precise when you use which data. In particular, some datasets are 13C measurements while others are pFBA predictions and it is sometimes unclear which you use. I am also unclear why these two data should have the same status, as pFBA ‘only’ provides predictions, while 13C are actual measurements.

2) It was unclear to me how exactly the covariates for the logistic regression were chosen. E.g., with 100 possible metabolites you would already have ~160.000 models with exactly three covariates to choose from.

3) For the covariates chosen in the end: how clear-cut was this decision. Are there other sets of metabolites that would provide similar fit and predictive accuracy? Did you check which metabolites your method chose and if they make biological sense as regulators for a specific enzymatic reaction?

4) How exactly did you perform the cross-validation? Your dataset contains duplicates of the same conditions, so arbitrary splitting will likely produce training and test sets that contain one duplicate each of the same condition. This would not provide a valid error estimate.

5) I did not understand the reasoning for your subsystem analysis. What can we learn from this regarding the accuracy of your method?

6) For the subsystems: You use a zero-one criterion effectively counting how many values fall within a prediction interval. But that would mean that a very bad model with very wide prediction interval will perform excellently according to your criterion.

7) Similarly, why do you choose the correlation coefficient as a measure of predictive power? And why do you choose different cut-offs for a ‘good’ coefficient (0.8 sometimes, other times 0.6).

8) With your various analyses I still remain somewhat unconvinced of the power of your method. One good comparison would be of the flux prediction with and without the ‘eta-correction’ from your method. How much do fluxes actually change when accounting for the metabolic flux effects?

9) Please provide some information about the distribution of ‘eta’ over conditions and reactions/enzymes. Does this function actually contribute to modulating the flux? Or are the values close to one anyway?

10) Your two studies use KO data for E. coli and growth conditions for yeast. These are biologically very different conditions. What is the reasoning that these are treated as equivalent?

Minor:

1) There is no mention of alternative methods, but much has been published on integrating and using different -omics data for metabolic models. Please provide some context how your method fits into this landscape.

2) How is your method related to ‘model balancing’ (if at all)? This seems to address a very similar problem.

3) You discuss ‘metabolic concentration effects’ throughout the manuscript and use the term in the title. But a somewhat vague definition only occurs in the method section after page 10. Please provide some definition or at least description of what you mean early on, as this concept is central to your method.

4) What were the growth conditions in your dataset? Are they comparable or do they include different carbon sources, for example?

5) Your method hinges a lot on the maximal apparent k_cat, which you estimate as the maximal observed value in the data. Why is this a useful estimate? Could one improve the method by considering the distribution of this maximum?

Reviewer #3: In this work, the authors develop a hybrid machine learning and constraint-based metabolic modeling method termed KineFlux to integrate enzyme cost considerations with metabolic concentration effects. They attempt to model the k_apparent parameter, which is an outcome of metabolite concentration effects, by obtaining a relationship between this parameter and a metabolic concentration proxy obtained by computed flux-sums. The authors then use the inferred relationship to design a constraint-based metabolic modeling approach to incorporate the modeled metabolite concentration effects, thereby (hopefully) obtaining a more accurate flux simulation. It’s an interesting and clever method that seems to have conceptual merit.

The practical question is given the series of sequential approximations necessary to implement the approach, how well does it work in the end? My initial questions were 1) whether there is sufficient proteomics data to accurately train the estimator model, 2) whether flux-sums are sufficiently accurate proxies for metabolite concentrations, and 3) whether pFBA is a sufficient method for accurately estimating flux (especially given that the point of the developed method is seemingly to improve on pFBA predictions). The answers to these questions primarily depend on the final performance of the method, as it is difficult to determine their answers independently given lack of data on each component.

My concerns with the work as presented are primarily 1) a lack of controls to which the method’s performance can be compared (especially given that pFBA is often quite good at estimating fluxes), and 2) a lack of detailed investigation of the metabolite effects, in terms of presenting overall characteristics of the estimated metabolite effects as well as specific case studies that could build confidence in the method’s predictions.

I hope that the authors can expand on their work to build confidence that the predictions made by the method are meaningful. My specific comments are below.

Major Comments

- Primary issue 1: Lack of control/reference comparisons. The authors produce a variety of parity plots throughout the manuscript comparing their model’s predictions to ‘data’ (pFBA flux simulations?), which they claim is evidence of improved predictions. However, in Figure 2 for example, it is not clear to me if the authors are making a completely independent prediction or just recovering the relationships the model was trained on. It would be good to compare this to a control, such as logit models trained on permuted data for example, as well as examining performance using holdout data. It also seems like these predictions should be compared to uncorrected pFBA-based flux predictions, and predictions with random etas assigned within an expected range, for example.

- Primary issue 2: Lack of details on results. The authors present a variety of parity plots but very little detailed discussion of 1) what the intermediate steps look like (for example the distribution of metabolite effects (eta), or some description of what the logit models look like on a per-reaction basis – how much do they change? What range of values is expected?) and 2) specific reactions exhibiting significant metabolite concentration effects, such as the MDH as discussed by Milo and colleagues in their original paper describing k_apparent_max (PMID: 26951675).

- Issue 3: It is not clear to me which “v_data” values are from 13C MFA vs pFBA. In line 208, the authors state “Second, the discrepancy may arise from the differing sources of flux estimates: the flux estimates in the unseen condition are based on pFBA, whereas the training data relies on more accurate 13C MFA.” In Figure 5, which is the unseen conditions example, the axes still say v_data and not v_pFBA or similar. The axes should be clearly labeled throughout with the source of the flux values, whether C13 MFA or pFBA.

- Issue 4: I am concerned about the subsystem analysis in Figure 3B. It seems that the model claims to predict fluxes well in peripheral metabolism but performs notably more poorly in central metabolism (glycolysis, pentose phosphate pathway, TCA cycle). Typically peripheral metabolism is tightly constrained by growth rate in these models while central metabolism has more flexibility. I can’t help but wonder if these statistics are inflated by hard constraints in the model. Can the authors please add appropriate control comparisons, such as the uncorrected pFBA-based flux predictions, and predictions with random etas assigned within an expected range, for example?

Minor Comments

- Line 49 “the flux predictions resulting from enzyme-constrained models are obtained under the implicit assumption of saturation of enzymes by their substrates” – this is not true (I can’t speak for all methods, but many do not make this assumption). A keff is assumed at some level of saturation that is assumed condition invariant, but not necessarily saturated.

- Line 63 – No need for flowery but empty adjectives like an ‘innovative combination’.

- Line 83: “Fluxomic data can be obtained by applying parsimonious flux balance analysis (pFBA)” – pFBA does not provide “fluxomic data” but rather “fluxomic predictions”

- Line 93 and the corresponding paragraph – “flux-sums” (defined in the caption of Figure 1 as the sum of fluxes producing a metabolite) are not adequately introduced for the reader to understand this section. A longer discussion of the previous work here and the principles underlying the connection between flux-sums and metabolite concentrations would be warranted here to justify the approach.

- For many of the scatter plots throughout the paper (e.g. Figure 5B), adding labels for at least a readable portion of non-trivial off diagonal elements would be a big improvement, so that the reader can better understand where the model has inaccuracies.

**Have the authors made all data and (if applicable) computational code underlying the findings in their manuscript fully available?**

Reviewer #1: None

Reviewer #2: Yes

Reviewer #3: Yes

PLOS authors have the option to publish the peer review history of their article (what does this mean? ). If published, this will include your full peer review and any attached files.

**Do you want your identity to be public for this peer review?** For information about this choice, including consent withdrawal, please see our Privacy Policy .

Reviewer #1: No

Reviewer #2: No

Reviewer #3: No

**Figure resubmission:**
---

## [Decision Letter · Decision Letter 1]

9 Jan 2026

PCOMPBIOL-D-25-01676R1

Accurate prediction of flux distributions compatible with metabolite concentration effects in genome-scale metabolic networks

PLOS Computational Biology

Dear Zoran,

My best wishes for a Healthy, Happy, and Prosperous New Year!

Vassily

Dear Dr. Nikoloski,

Thank you for submitting your manuscript to PLOS Computational Biology. After careful consideration, we feel that it has merit but does not fully meet PLOS Computational Biology's publication criteria as it currently stands. Therefore, we invite you to submit a revised version of the manuscript that addresses the points raised during the review process.

We look forward to receiving your revised manuscript.

Kind regards,

Vassily Hatzimanikatis

Academic Editor

PLOS Computational Biology

Pedro Mendes

Section Editor

PLOS Computational Biology

**Journal Requirements:**

- Please amend your detailed Financial Disclosure statement. This is published with the article. It must therefore be completed in full sentences and contain the exact wording you wish to be published.

State what role the funders took in the study. If the funders had no role in your study, please state: "The funders had no role in study design, data collection and analysis, decision to publish, or preparation of the manuscript.".

**Reviewers' comments:**

Reviewer's Responses to Questions

**Comments to the Authors:**

Reviewer #1: While the authors responded well to most of my previous comments, I do not think that they adequately addressed the following comment:

“Although the authors mention that KineFlux utilizes enzyme-constrained models, they do

not detail how the framework integrates such constraints. The Methods section lacks

information on the construction of enzyme-constrained models for E. coli and S.

cerevisiae. Additionally, it would be valuable to clarify whether KineFlux can be applied

to classical metabolic models without enzyme constraints.”

This comment requested details on model construction and discussion on applicability of KineFlux to classical metabolic models. However, the authors provided an irrelevant response.

Reviewer #2: I thank the authors for the improvements in their manuscript. In particular the use of data sets, the cross-validation procedures, and the explanation of the method as such are now much clearer. I also appreciate the changes to the Introduction, which now more clearly states the aims and central ideas of the manuscript.

Despite the many improvements, I believe that several major issues have not yet been addressed appropriately in this revision.

1) What exactly is the proposed use-case for this method? You argue that the method consistently gives predictions close to the pFBA predictions. So why to use the established and trusted pFBA/FBA in the first place? What cases does the proposed method cover that the current methods do not?

2) Similar to the other reviewers, I believe the method is interesting and has merit, but that the claim that it captures metabolite concentration effects has not been sufficiently demonstrated. If true, then the method would add a very valuable perspective on regulation in genome-scale metabolic networks. However, your comparisons with BRENDA and STITCH seem to indicate that the method almost never picks known regulatory metabolites for the logistic regression model. Could you expand on the use of your method for identifying key regulators of reactions and/or reactions that are highly regulated (in specific conditions)? As a minor point: why do you use additional metabolites _not_ identified as regulators in your logistic regression for comparison to the databases?

3a) I maintain my previously comment that both the Pearson correlation coefficient is inadequate for evaluation comparing KineFlux and (p)FBA prediction, at least as reported. Clearly, you can get a correlation of one with _any_ linear relationship between estimate and prediction, which says nothing about the accuracy of the prediction. For example, Fig 2B clearly shows consistent underestimation of fluxes by KineFlux yet yields an excellent correlation.

3b) Similarly, how do you consider a median R^2 of 50% for your models as proof that flux-sums are reliable features?

3c) I also maintain that your performance evaluation based on the prediction interval is not entirely meaningful. If your prediction model is bad, this evaluation will show excellent performance. And the prediction interval shown in Fig 3A spans four orders of magnitude. How is a prediction within such a range successful?

4) I am a bit puzzled by the distribution of the eta-values, which you provide as roughly 0.5 with not-too large standard deviation. Does that mean that—according to your method—the vast majority of reaction operate at about 50% of their maximal capacity? Is this realistic? Could you provide a plot of the probability density of eta for some scenarios?

Minor comment:

- Fig 3A shows a substantial number of fluxes with either extremely low (zero?) estimate or prediction. Were this excluded from the evaluation and the calculation of the correlation coefficient?

Reviewer #3: The authors have sufficiently addressed my previous concerns.

One remaining minor issue - the authors changed Line 27 to "that all enzymes are saturated to the same level," but this still isn't quite correct. The assumption is that each enzyme is saturated to a constant level, which may be enzyme specific but is condition invariant. Thus, v/vmax for enzyme 1 may differ from that for enzyme 2, but v/vmax does not change for enzyme 1 across conditions (a weakness of current methods for enzyme-constrained modeling). Just a minor phrasing suggestion.

**Have the authors made all data and (if applicable) computational code underlying the findings in their manuscript fully available?**

Reviewer #1: None

Reviewer #2: Yes

Reviewer #3: Yes

PLOS authors have the option to publish the peer review history of their article (what does this mean?). If published, this will include your full peer review and any attached files.

**Do you want your identity to be public for this peer review?** For information about this choice, including consent withdrawal, please see our Privacy Policy .

Reviewer #1: No

Reviewer #2: No

Reviewer #3: No

**Figure resubmission:**
---

## [Decision Letter · Decision Letter 2]

26 Feb 2026

Dear Zoran,

We are pleased to inform you that your manuscript 'Accurate prediction of flux distributions compatible with metabolite concentration effects in genome-scale metabolic networks' has been provisionally accepted for publication in PLOS Computational Biology.

Congratulations on the acceptance of your paper!

Reviewer 2 has provided a few minor comments.

Although these do not affect the acceptance decision,

I would recommend that you consider addressing them,

or briefly respond to them in your revision.

This is not a condition for acceptance,

but it would further strengthen the final version of the manuscript.

Best regards,

Vassily Hatzimanikatis

Academic Editor

PLOS Computational Biology

Pedro Mendes

Section Editor

PLOS Computational Biology

Reviewer's Responses to Questions

**Comments to the Authors:**

Reviewer #1: I do not have any comment.

Reviewer #2: I thank the authors for the further clarifications provided in the rebuttal letter and in the updated manuscript. Most of my previous comments are now addressed. However, two points of remain open.

1) The issue of using a prediction interval to evaluate model performance is still not sufficiently addressed.

The evaluation in the paragraph starting line 179 is based on the notion of ‘well-prediction’ where a flux is well-predicted if its value falls within the prediction interval of the trained model. But—regardless of the quality of the model—if the prediction intervals are well-calibrated in the statistical sense, then we expect that 90% of the measured fluxes fall within a 90% prediction interval. This holds for ‘good’ and ‘bad’ models, the difference being that ‘bad’ models likely have wider intervals. Counting the number of fluxed falling into the interval is therefore not a measure of prediction quality.

Could you please explain how this measure provides an evaluation of model quality? And why this measure is required or useful, given that model prediction quality is already assessed by a range of measures in your Table 1?

2) The next section claims that ‘KineFlux points at enzyme regulators’, but the number of known regulators the method identifies is very low, even when additionally considering all metabolites that are highly correlated with the ones selected by the method. Please either show that this number is much larger than expected by chance and/or revise the statements and the section title. Please also clarify the relation between biological effectors (as mentioned in lines 32 and 87-89) and the ‘metabolic concentration effects’ used in your method. Clearly, KineFlux can be highly predictive even if not selecting known regulators, but the current manuscript is slightly ambiguous about this.

**Have the authors made all data and (if applicable) computational code underlying the findings in their manuscript fully available?**

Reviewer #1: None

Reviewer #2: Yes

PLOS authors have the option to publish the peer review history of their article (what does this mean? ). If published, this will include your full peer review and any attached files.

**Do you want your identity to be public for this peer review?** For information about this choice, including consent withdrawal, please see our Privacy Policy .

Reviewer #1: No

Reviewer #2: No

---

## [Editor Report · Acceptance letter]

PCOMPBIOL-D-25-01676R2

Accurate prediction of flux distributions compatible with metabolite concentration effects in genome-scale metabolic networks

Dear Dr Nikoloski,

I am pleased to inform you that your manuscript has been formally accepted for publication in PLOS Computational Biology. Your manuscript is now with our production department and you will be notified of the publication date in due course.

With kind regards,

Anita Estes
